# Dielectric Barrier Discharge Plasma-Assisted Preparation of Chitosan-Based Hydrogels

**DOI:** 10.3390/ijms25042418

**Published:** 2024-02-19

**Authors:** Runing Liang, Dan Zhang, Junwei Guo, Shaohuang Bian, Cheng Yang, Lusi A, Weiwei Zhang, Feng Huang

**Affiliations:** 1College of Science, China Agricultural University, Beijing 100083, China; liangruning@cau.edu.cn (R.L.); zhangddd@cau.edu.cn (D.Z.); guojw020@cau.edu.cn (J.G.); ycylmchy@163.com (C.Y.); wwzhang@cau.edu.cn (W.Z.); 2College of Information and Electrical Engineering, China Agricultural University, Beijing 100083, China; b20233080721@cau.edu.com (S.B.); alos@cau.edu.cn (L.A.)

**Keywords:** low temperature plasma, dielectric barrier discharge, chitosan-based hydrogels, preparation, characteristics

## Abstract

Chitosan is widely used in the production of various hydrogels due to its non-biological toxicity, good biocompatibility, and strong biodegradability. However, chitosan-based hydrogels have not been widely used in tissue engineering due to their poor mechanical strength, poor stability and high biotoxicity of cross-linking agents. As a green technology, low temperature plasma is rich in active groups that can be involved in various chemical reactions, such as replacing the components on the chitosan chain, contributing to the cross-linking of chitosan. In this study, a plasma-assisted preparation method of chitosan-based hydrogels was developed and the properties, including mechanics, water absorption, and degradation (or stability), were characterized and analyzed. It is proved that plasma treatment plays a significant role in improving the mechanical strength and stability of hydrogels.

## 1. Introduction

Chitosan, a product of deacetylation of chitin, is a natural cationic polysaccharide compound, which contains positively charged free amino groups [1]. With the strong chemical activity, it has functions of enhancing immunity, antibacterial, anti-aging, preventing diseases, promoting healing, and regulating physiology [2]. In addition, it is environmentally friendly, highly biodegradable, easily obtained, making it a green, natural-source polymer material [3]. Chitosan is promising for the preparation of novel hydrogels, which are widely used in fields of medicine [4], drug delivery [5], and tissue engineering [6], especially the superiority in the field of wound healing [7,8]. Chitosan-gluconic acid conjugate hydrogels [9] under autoclaving (121 °C, 20 min) simultaneously achieved gelation of the solution and sterilization of the hydrogel, which retained favorable biological properties and was promising as wound dressings. Chitosan hydrogels can be linked to biosensor physical elements for disease diagnosis and treatment. Hydrogels formed by cross-linking chitosan with a mixture of graphene oxide and dopamine have strong adhesion, self-healing ability, and good electrical conductivity [10], which can be applied to wearable sensors to detect human activities. Chitosan can also be used as a moisturizing ingredient in cosmetics. Chitosan, compounded with gellan gum [11], has strong water retention properties and can effectively retain water for about 9 h. In addition, chitosan can also be used in environmental protection, which widely uses sustainable adsorbents for wastewater remediation [12], contaminated soil [13], and dye removal [14]. For example, in the dye adsorption experiment using CO_2_-activated chitosan [14], the crosslinking of the amino group of chitosan and carbonic acid form carbamate at high temperatures, resulting in a protonated amino group (·${NH}_{3}^{+}$) which enhances the adsorption capacity of dyes, providing a highly efficient and clean treatment method for wastewater containing dyes.

Common chitosan hydrogels are mainly chemically cross-linked using small molecule cross-linking agents, such as glutaraldehyde [15], which are toxic and prone to causing pollution. Therefore, exploring non-toxic and non-polluting cross-linking methods has become the focus of chitosan hydrogel research. In recent years, sodium alginate, as a natural anionic polysaccharide compound [16], contains carboxyl groups on its molecular chain, which is easy to cross-link with other positively charged cations or polycations to form a hydrogel [17]. Therefore, it can interact with positively charged amino groups on chitosan to cross-link and form a hydrogel. Meanwhile, due to the good biocompatibility of sodium alginate, it currently has gradually replaced the chemical cross-linking agent to cross-link with chitosan to form hydrogels [18]. Some studies have shown that the addition of alginate can improve the poor mechanical properties of chitosan hydrogels [19].

Plasma is an ionized gas which contains a large number of electrons and ions that are macroscopically electrically neutral [20]. Low temperature plasma is a green and non-polluting technology, which contains a large number of ions, electrons, and un-ionized neutral particles, as well as a variety of free radicals and other active components, accompanied by ultraviolet radiation and so on. The contact of highly active components in plasma with polymers can initiate various chemical reactions [21]. The interaction of plasma and polymers may lead to different processes, such as functionalization, cross-linking, and etching [22], which provide the feasibility of plasma application in hydrogel polymer preparation. Figure 1 shows a schematic diagram of plasma-liquid interaction [23]. When a liquid sample is exposed to atmospheric pressure plasma, reactive species such as Reactive Oxygen Species (ROS) and Reactive Nitrogen Species (RNS) formed in the plasma interact with the liquid through collision, diffusion, absorption, and chemical transfer [24], and then can react chemically with the polymers in the liquid [25]. The newly formed radicals at the plasma-liquid interface will tend to remain in the native liquid phase to initiate post-discharge reactions within the liquid. Excess unreacted and unbound electrons may also generate additional reactions in the liquid phase.

Although there have been reports related to plasma preparation and the treatment of chitosan hydrogels [26,27], in these studies the concentration of the active substance on the surface of the solution is much higher than that in the depth of the solution due to the plasma treatment process, although there is an osmotic effect. At the same time, the local thermal effect on the surface is obvious, the water is more likely to evaporate, and polymerizations of the cross-linked network are more likely to occur. The formation of the cross-linked network prevents the active substances from entering the depth of the solution. In the meantime, due to the evaporation of water caused by the thermal effect of the reaction or the increase of the processing time, the hydrogel film already formed on the surface may break, which is more likely to lead to the degradation of the chitosan molecules so that it is not possible to further react and successfully prepare a block hydrogel. Therefore, the current plasma researches in chitosan hydrogels mainly focus on the preparation of chitosan hydrogel films by plasma or the modification on the chitosan-based hydrogel surface by plasma treatment [28,29,30].

With the deepening of research and technological advances, chitosan-based hydrogels have been the current trending materials for smart hydrogel research. However, the existing chitosan hydrogels still have problems, such as poor mechanical strength, unstable performance, sensitivity to the environment, and prone to in vivo degradation, which affect its possible applications, for example, in applications as a tissue support material. Meanwhile, there is still much space for progress in biocompatibility.

In order to prepare chitosan-based hydrogels with stable structures, good mechanics, and biocompatibility, a plasma-assisted method was used. The traditional method of acid dissolution of chitosan produces many free amino groups. The chitosan chain is more easily attacked by active substances, resulting in chitosan chains breaking and degrading without cross-linking. In this experiment, direct dielectric barrier discharge (DBD) plasma treatment was used to dissolve the chitosan suspension. During the process of treatment, the active substances produced in plasma can effectively combine with chitosan to dissolve chitosan. Some of the active groups can replace the corresponding bond positions on chitosan during the treatment process, increasing the sites for chitosan to cross-link afterwards. After that, the use of crosslinking agents or physical crosslinking is more effective than the traditional use of acetic acid to directly dissolve chitosan.

In this paper, the chitosan (CTS) hydrogels, sodium alginate (SA) hydrogels and chitosan-sodium alginate (CTS + SA) composite hydrogels were prepared by conventional methods and plasma-assisted hydrogels were compared. The mechanical properties and degradation properties of the hydrogels were characterized, and the effects of plasma assistance on the structures and properties of the hydrogels were investigated.

## 2. Results and Discussion

### 2.1. Plasma-Assisted Dissolution of Chitosan under Different Treatment Voltages and Time

Figure 2 shows the chitosan suspension and its temperature change during plasma treatment with different voltages and time. Figure 2a shows the chitosan suspension without plasma treatment, indicating that chitosan is usually insoluble in water and uniformly dispersed in water, forming a suspension. At the same discharge voltage, for example, at 45 kV, chitosan gradually dissolves as the treatment time increases. However, there is a small amount of undissolved chitosan when the treatment time is 15 min. All chitosan was dissolved when the treatment time reaches 18 min. When the discharge voltage was 60 kV, chitosan was completely dissolved by the treatment of 15 min. When the discharge voltage was 75 kV, some chitosan was charred and black flocculent appeared in the solution at 12 min. Figure 2b shows the side view and top view of chitosan dissolved at different discharge voltages, which indicates that chitosan was dissolved best at the treatment of 60 kV and 15 min.

During plasma treatment, the excessive heat accumulated may lead to the chitosan coking, which will affect the dissolution effect. Intermittent plasma treatment was used, for example, 3 min of plasma treatment and then 2 min of cooling, and 5 cycles with 15 min for the total treatment time and 25 min for both treatment and cooling in total. Figure 2c shows the temperature changes during treatment at different voltages. It can be seen that the higher discharge voltage, the higher temperature it caused, and after 25 min of intermittent treatment and cooling, the hydrogel was cooled to room temperature for the subsequent structural measurements and performance characterization.

The dissolution of chitosan by acetic acid is due to the combination of H^+^ dissociated from acetic acid and the amino group on chitosan, forming ${NH}_{4}^{+}$, which causes the destruction of the original hydrogen bond of chitosan and dissolution. The dissolution of the plasma-treated chitosan suspension can be understood as a large number of active substances, free cations, and anions, which are formed in the liquid under the condition of plasma treatment, in which the free anions can form hydrogen bonds with both the hydrogen atoms in the hydroxyl group of the chitosan macromolecular chain and the hydrogen atoms in the amino group of the macromolecular chain, while the free cations interact with the oxygen in the chitosan macromolecules that has lost hydrogen atoms. Thus, the original hydrogen bond of chitosan is broken, resulting in the dissolution of chitosan in ionic liquid. Plasma can dissolve chitosan, which is also due to the plasma discharge process generating a large number of reactive groups (HNO_2_, HNO_3_, and so on), which can react with water molecules and generate H^+^ through a series of reactions, forming the acidic environment [31]. Chitosan dispersed in water will gradually dissolve in the acidic environment generated by the reaction between plasma and water. It was reported that when treated for a certain amount of time, chitosan viscosity decreased and chitosan degradation occurred [32]. Chitosan dissolution takes a longer time under lower voltage treatment, and chitosan solution produces flocculent under higher voltage treatment, which indicates the chitosan dissolution should be under appropriate discharge parameters, which provides key parameters for the preparation of chitosan-based hydrogels.

It should be noted that plasma treatment of chitosan suspension can promote the dissolution of chitosan, but the already dissolved chitosan will still be attacked by the active substances generated by the plasma. So, it is necessary to control the treatment parameters including voltage and time, and plasma treatment should be stopped in time after the chitosan is dissolved. If the molecular chains produced during the reaction are lower than 3200 Da, the chitosan will become chitooligo saccharide and the molecular chains will not be enough to form a network structure, which will affect the cross-linking effect and will have a negative effect on the quality and physicochemical properties of hydrogels.

### 2.2. Active Substance-Assisted Dissolution of Chitosan in Plasma

During the treatment of the chitosan solution in plasma, the charge and active components in the plasma interact with water to produce plasma-activated water with active substances such as ·OH, H_2_O_2_, and $O_{2}^{-}$ [33]. On the one hand, chitosan can be dissolved in the acidic condition formed by the plasma-activated water. On the other hand, these active substances can oxidize and degrade part of the chitosan under the acidic conditions of the plasma-activated water, improving the dissolution effect of chitosan [34]. In the absence of chitosan, OH radicals generated by plasma burst to form a variety of substances, including H_2_O_2_. The reaction ·OH + ·OH → H_2_O_2_ has been shown to be the main chemical pathway for the formation of hydrogen peroxide in plasma-treated solutions [32]. Therefore, the rate of OH radical production can be qualitatively and indirectly shown by measuring the H_2_O_2_ concentration of the treated chitosan solution.

The changes of hydrogen peroxide and superoxide concentration with treatment time during plasma treatment of deionized water are shown in Figure 3. From Figure 3a, it can be seen that with the increase of the discharge time, H_2_O_2_ produced in the solution gradually increases, which can effectively attack the β-1,4 glucosidic bond in chitosan macromolecules in the aqueous solution [35], breaking the long chain of chitosan and causing partial depolymerization to occur. The concentration of $O_{2}^{-}$ in Figure 3b decreases with the increase of treatment time, showing a large amount of $O_{2}^{-}$ in the solution to react with water to form H_2_O_2_ [36] through the reaction equation $O_{2}^{-}$ + H_2_O + e^−^ → H_2_O_2_.

Based on previous literature [37,38], the reaction mechanism of ·OH with chitosan during plasma treatment can be deduced. During plasma treatment, hydroxyl radicals were formed in the solution, which attacked the C–O bond of chitosan molecules and broke the glycosidic bond to form low molecular mass chitosan or broke the pyranose ring to form the carbonyl group or change the conformation of chitosan from β to α. These possible chemical pathways of chitosan modification by hydroxyl radicals generated during plasma treatment of solutions in atmospheric pressure are shown in Figure 4. At the same time, plasma reactive substances may interact with chitosan, leading to surface activation, formation of free radicals and functional groups, and a small part of chitosan molecules becoming bound by covalent bonds [32].

According to previous studies [39], the plasma-activated water was only able to dissolve a small portion of chitosan, which could not meet the requirement for the preparation of bulk chitosan with high mechanical properties. The precipitated chitosan needed to be further centrifuged to separate, which increased the difficulty of the experiment. Hence, in our experiment, direct plasma treatment rather than plasma-activated water was chosen to dissolve the chitosan suspension. The reason for this may be due to the fact that hydroxyl radical, which plays the major role in chitosan dissolution, has a short lifetime [40] and if water is treated first before dissolving chitosan, chitosan cannot be completely dissolved, i.e., hydroxyl radical cannot react with chitosan molecules in time. In the process of chitosan dissolution treated by plasma, there are also various forms such as high temperature degradation, ozone oxidation, supercritical water oxidation, ultraviolet photolysis, high-energy electron bombardment, excited particles, and so on, accompanied by physical effects such as ultraviolet radiation, local high temperatures, and high-energy shock waves generated by chemical effects. Their roles in plasma treatment need to be further studied [41].

The traditional method of acid dissolution of chitosan produces many free amino groups, and the chitosan chain is more easily attacked by active substances, resulting in the chitosan chain breaking too much, degrading, and being difficult to cross-link. While using plasma treatment on chitosan suspension for dissolution, the active substance can effectively combine with chitosan to dissolve chitosan. Some of the active groups can replace the corresponding positions on chitosan during the treatment process, increasing the sites for chitosan to cross-link afterwards, which is more effective than the traditional acid dissolution methods.

### 2.3. Plasma Preparation of Different Types of Hydrogels

In this experiment, we compare three kinds of hydrogels, whose names and preparation parameters are shown in Table 1. In each kind of hydrogels, there are two preparation methods, in which the structures and properties with and without plasma assistance will be compared, including surface morphology, infrared absorption spectrum, degradation properties, and so on.

Figure 5 shows the comparison of three kinds of plasma-treated chitosan, sodium alginate, chitosan-sodium alginate composite hydrogels, and their freeze-dried series. It can be seen that the colors of plasma-treated hydrogels and their freeze-drying series were darker than those without plasma treatment, indicating the significant effect of plasma in promoting the dissolution and cross-linking of chitosan and sodium alginate. In addition, the hydrogels prepared by the plasma had rougher surfaces after freeze-drying. In addition, the subfigure of Figure 5a shows the prepared hydrogel with plasma-assistance is cylindrical shape with the diameter of 3.5 cm and height of 1.5 cm, indicating that the size of the prepared bulk hydrogels in this experiment is much larger than that reported in the literature [42,43,44], in which the obtained hydrogels were mainly millimeter sized hydrogels or the obtained hydrogels were only cast film or flocculent. It is due to that plasma treatment can promote electrostatic interactions between the positively and negatively charged ions and cross-linking to form the hydrogels with a better network structure.

### 2.4. Surface Morphology of Different Types of Hydrogels Prepared by Plasma

The cross-linking of hydrogels can also be indirectly shown from the surface morphology. Figure 6 shows the scanning electron microscopy (SEM) images of the freeze-dried hydrogels, including chitosan hydrogel, sodium alginate hydrogel, and chitosan-sodium alginate composite hydrogel. For a certain hydrogel, comparison of the morphologies with and without treatment shows that the pore size of the hydrogel with plasma assistance preparation treatment becomes smaller, indicating that the plasma treatment can increase the cross-linking density. It is due to that plasma treatment can produce shorter molecular chains, which can increase the binding sites and improve the cross-linking during the subsequent cross-linking process of the hydrogels. In addition, plasma treatment can produce more anions and cations in the solution and accordingly increase the electrostatic interactions during cross-linking process. In addition, the network structure of the plasma prepared chitosan-sodium alginate composite hydrogel is denser than both the plasma treated chitosan hydrogel and the plasma treated sodium alginate hydrogel, and has the smallest mean pore size (about 100 μm). These phenomena indicate the composite hydrogel has stronger cross-linking and better mechanical property.

### 2.5. Infrared Spectra of Different Kinds of Hydrogels Prepared by Plasma

The composition of a hydrogel and the bonding of its internal groups can be detected by Fourier Transform Infrared Spectroscopy (FTIR). When a beam of infrared light with a continuous wavelength passes through a substance, a group in the substance molecule vibrates or rotates at the same frequency as the infrared light, the molecule absorbs energy and jumps from the original ground state energy level to a higher energy level. When cross-linking occurs in hydrogel, the molecular chains are bonded or interact with each other. When infrared light passes through the hydrogel, the light at the corresponding wavelength is absorbed by the chemical bond, while the light at other wavelengths has no effect. This makes it possible to detect changes in the composition of the hydrogel and the bonding of the internal groups. As shown in Figure 7, the infrared spectrum of freeze-dried hydrogel powders after compaction are compared. It can be seen that in the FTIR spectrum of chitosan, the absorption peaks at 1540 cm^−1^ and 1410 cm^−1^ are from the amide bands [45] and 1020 cm^−1^ are from ammonium ions. After being plasma treated, the peaks of amide bands nearly disappear and the peak of ammonium ions significantly weakens. In the spectrum of sodium alginate gels, the peaks at 1596 cm^−1^ are the characteristic absorption peaks of sodium alginate [46], and in this spectrum the peak of 1020 cm^−1^ is slightly weakened after plasma treatment. Through comparing the spectrum a, c and e, it can be seen the significantly weakened peaks at 1596, 1540, 1410, and 1020 cm^−1^ indicate the interaction of ammonium ions on the molecular chain of chitosan and the carboxylate ions on the molecular chain of sodium alginate. In addition, from the comparison of the spectrum e and f, when applying plasma treatment, there is a broad absorption peak at about 600 cm^−1^, which corresponds to the product after the cross-linking reaction of chitosan and sodium alginate.

### 2.6. Mechanical Properties of Different Types of Hydrogels Prepared by Plasma

Mechanical properties are critical to the practical application of hydrogels. Hydrogels of natural polymers are usually fragile. Figure 8 shows the stress–strain curves and compression modulus (calculated from the ratio of compression force and compression strain in the stress–strain curves) obtained from the compression tests of those hydrogels in the axial direction. It can be seen that the composite hydrogel has significantly higher mechanical strength compared to the hydrogels of only chitosan or sodium alginate. That is, when the strain is the same, the stress applied on the composite hydrogel is greater, shown in Figure 8a. In other words, the mechanical properties of the composite hydrogel are significantly better than single component hydrogels, which is consistent with the results in SEM diagrams (Figure 6). This is probably due to the following reason. When ionic cross-linking occurs between chitosan and sodium alginate, there are more groups on their molecular chains that can be involved in the reaction, and the reaction is more adequate. In addition, more opposite attraction charges occur between them, forming a denser cross-linked network, and thus, its mechanical strength is higher. Figure 8b shows that the compressive modulus of chitosan and sodium alginate composite hydrogel (8.08 ± 0.14 kPa) is significantly greater than that of chitosan hydrogel (3.22 ± 0.52 kPa) and sodium alginate hydrogel (3.85 ± 0.30 kPa). In addition, all hydrogels with plasma assistance have a higher compression modulus than those without plasma treatment. For example, the compressive modulus of chitosan hydrogel, sodium alginate hydrogel, and their composite hydrogel with plasma-assisted preparation increase by 19.57%, 15.50% and 8.39%, respectively. Therefore, plasma treatment can improve the mechanical strength and stability of hydrogels. Similar to the results of a previous study [47], the mechanical strength of all kinds of hydrogels were further improved after plasma treatment. Plasma improvement is mainly due to the evidence that plasma treatment makes chitosan macromolecules partially hydrolyzed, generating more charges and radicals for reaction cross-linking, and short-chain chitosan is more likely to react with other substances than long-chain chitosan, making the reaction more adequate. At the same time, the plasma-treated sodium alginate molecular chain with more polarized radicals also contributes to the improvement of mechanical strength.

### 2.7. Porosity of Different Types of Hydrogels Prepared by Plasma

Porosity is the percentage of pores in a hydrogel in relation to its total volume. The porosity of a hydrogel can visually indicate the water absorption and retention properties of the hydrogel. The porosity and size of pores also affect the strength and stability of hydrogels. The porosities of these hydrogels are shown in Figure 9, where it can be seen that the porosities of these hydrogels are between 93% and 97%. It can be seen that the porosity of plasma-assisted hydrogels is slightly lower than those without applying plasma. This is because plasma treatment increases the degree of cross-linking of the hydrogels, and accordingly decreases the porosity. The increased porosity of chitosan will help the hydrogel to absorb more water (conducive to cell growth). However, too high porosity will reduce the mechanical strength of hydrogels. At present, mechanical strength is the main problem of chitosan-based hydrogels encountered in application. So, the appropriate reduction of the porosity is favorable for the improvement of the mechanical strength. The results of this experiment show that the plasma-assisted preparation of hydrogels can properly reduce porosity and improve the performance in mechanical strength (as shown in Section 2.6).

### 2.8. Water Absorption and Swelling Ratio of Different Types of Hydrogels Prepared by Plasma

Water absorption and the swelling ratio are the two key parameters that indicate the properties of hydrogels. The water absorption ratio means the mass of absorbed water divided by the total mass of hydrogel after absorbing water. The swelling ratio means the mass of the absorbed water divided by the original mass of freeze-dried hydrogel. The high swelling ratio of hydrogels is attributed to their porous and the abundance of hydrophilic groups in the polymer chains. Swelling process, i.e., swelling ratio versus time, are carried out on these hydrogels, which is shown in Figure 10. It can be seen that without applying plasma, the swelling process reaches equilibrium after 8 h for sodium alginate hydrogel and chitosan-sodium alginate composite hydrogel, while the swelling process of the chitosan hydrogel is still continuing to increase with time. Comparing the swelling process of hydrogels with and without applying plasma, the swelling trends with time is similar, but the swelling process curves of the plasma-assisted hydrogels are lower than those of without plasma assistance. This can be understood as follows. During the swelling process, water penetrates into the hydrogel to swell its volume, the three-dimensional network of the hydrogel extends to perform volume expansion in three-dimensions, and at the same time, the molecular network generates elastic forces that shrink the network. The swelling equilibrium position is reached when the two forces reach equilibrium. At this time, the swelling ratio of the hydrogel is related to the type of the hydrogel and the cross-linking density [48]. The higher the crosslink density, the lower the swelling ratio. From the swelling equilibrium point, it can be seen that the decrease of the swelling ratio of plasma-assisted hydrogel is due to the increase of the cross-linking density, which can be shown in Figure 10a. Figure 10c shows the water absorption of different hydrogels, from which it can be seen that the water absorptions of plasma-assisted hydrogels are very slightly reduced compared to those without plasma. The water absorption of a hydrogel is related to its porosity and the characteristics of its pores. This is because water is drawn in through the open pores of the hydrogels and penetrates into the interior through the connecting pores. The more and larger pores that are connected to the outside, the more water the pores absorb. From the perspective of application, when the water retention of a hydrogel reaches more than 80%, it can meet the requirements of application. On the contrary, too much water absorbed easily causes the reduction of mechanical strength of chitosan hydrogels and can make them easy to rupture, which is another main problem of chitosan hydrogels in the application of the current. Therefore, the appropriate reduction of water absorption is conducive to improving the mechanical strength. Combined with the SEM images in Figure 6, it can be found that the pore sizes of the hydrogels with plasma preparation are significantly reduced, indicating that plasma treatment can improve the crosslinking degree and reduce the pore sizes by modifying the groups on the molecular chain to appropriately reduce the water absorption ratio.

### 2.9. Degradation Properties of Different Types of Hydrogels Prepared by Plasma

Degradation ability is very important for the use of hydrogels in biomedical or tissue engineering fields. Shorter degradation time requires multiple replenishments or replacements, longer degradation time can affect the normal physiological functioning of the cell or organism, and appropriate degradation time can help to minimize any abnormal reactions that may exist. Degradation can be simulated using in vitro simulation, and the degradation degree can be characterized by the change of the compression modulus of hydrogels. Figure 11 shows the change of compression modulus of the hydrogels over time, in which E/E_0_ is the ratio of the compression modulus at that time to the initial compression modulus. As we can see, E/E_0_ gradually decreases with time, indicating that degradation is occurring. The degradation curve with time shows that E/E_0_ of plasma-assisted hydrogels is generally higher than those without plasma, i.e., the degradation ratio of plasma-assisted hydrogels is lower than those of without plasma, or the stability is higher than those of without plasma. In addition, E/E_0_ of the chitosan-sodium alginate composite hydrogel is significantly higher than that of single component hydrogels. Taking the composite hydrogel as an example, the initial compression modulus of hydrogel without plasma is 8.08 kPa and drops to about 3.24 kPa on the 16th day of degradation, which indicates that the hydrogel still retained about 40% of its original mechanical strength after 16 days of degradation. The initial compression modulus of the plasma-assisted composite hydrogel is 8.82 kPa and drops to 4.08 kPa on the 16th day, i.e., 46.28% of the compression modulus is retained, which indicates that the degradation ratio of the plasma-assisted composite hydrogel is reduced. That is, plasma-prepared hydrogel can enhance the anti-degradation ability.

### 2.10. Overall Analysis of Different Types of Hydrogels

Finally, the different properties, including mechanical properties (on the 16th day), porosity, water absorption ratio, swelling ratio, and degradation properties of the six groups of hydrogels are combined to evaluate the overall performance of the hydrogels. Through the effective multi-indicator evaluation method of TOPSIS analysis, the overall performances of the hydrogels (represented by the overall evaluation value f) are obtained. The larger the f-value, the better the overall evaluation performance of the hydrogel. Table 2 represents the characteristic values of the five properties and the overall evaluation value of each hydrogel. Table 2 shows that the mechanical strength and anti-degradation properties (E/E_0_) of hydrogels with plasma-assisted preparation are significantly improved with an increase within 9.2–19.6% and 15.2–64.3%, respectively. Meanwhile, the porosity, water absorption, and swelling performance are slightly decreased within 1.5–3.4%, 0.2–0.7% and 7.6–13.8%, respectively. From the comparison of the overall evaluation value *f*, it can be seen that the overall performance of plasma-assisted hydrogels is better than that of the hydrogels without using plasma. In addition, in these six groups of hydrogels, the composite hydrogel of chitosan and sodium alginate prepared by plasma-assisted method has the best comprehensive performance with the compression modulus of 8.82 kPa and 46.28% of the compression modulus is retained on the 16th day of the simulated degradation, showing the strong mechanical and anti-degradation properties.

## 3. Materials and Methods

### 3.1. Reagents

In this experiment, the materials used for the preparation, treatment and measurement of hydrogels are as follows. Chitosan (CAS 9012-76-4, (C_6_H_11_NO_4_)_n_, 310 to 375 kDa, degree of deacetylation > 80%) and sodium alginate (CAS 9005-3-3, (C_6_H_7_NaO_6_)_n_, (198)_n_, chemically pure) were provided by Sinopharm Chemical Reagent Co. (Shanghai, China). Acetic acid (CAS 64-19-7, ≥99.5%, pure for analysis) and glutaraldehyde (CAS 111-30-8, pure for analysis, 50% in H_2_O) was obtained from Aladdin Scientific Corp. (Shanghai, China). Anhydrous ethanol (CAS 64-17-5, ≥99.5%, pure for analysis) was sourced from Chinasun Specialty Products Co. (Changshu, China). Anhydrous calcium chloride (CAS 10043-52-4, pure for analysis) was supplied by Jiangsu Yatai Chemical Co. (Nantong, China). PBS buffer (XG3650) was obtained by Shenzhen Xigene Biotechnology Co. (Shenzhen, China).

### 3.2. Plasma Treatment Process

This experiment used a DBD device, schematically shown in Figure 12. The experimental device consists of two round aluminum plate electrodes. The lower electrode is grounded. During the treatment, a high borosilicate container covered with a quartz cover plate containing chitosan suspension, sodium alginate solution or their combination was placed between the upper and lower electrodes to be treated. Helium gas with a rate of 5 L/min was filled into the container at atmospheric pressure. The discharge current and voltage waveforms were measured by a Tektronix oscilloscope (Tektronix TBS1102B, Tektronix, Johnstown, PA, USA) and a voltage probe (Tektronix TPP0101, Tektronix, Johnstown, PA, USA). Then, the discharge power was calculated.

Figure 13 shows the discharge waveforms of discharge voltage and current at 60 kV acquired by a digital oscilloscope. It can be seen that the discharge voltage waveform applied to the electrodes is sinusoidal. The discharge state is between uniform discharge and filamentary AC discharge with one or more discharges in half a voltage cycle. The discharge power is calculated to be 150 W from the Lissajous plot of voltage-current curves. Figure 14 shows the image of discharge when helium was filled into the discharge chamber at atmospheric pressure, showing the purple dominant discharge colors.

### 3.3. Preparation of Plasma-Treated Hydrogels

#### 3.3.1. Preparation of Chitosan Hydrogels by Plasma Treatment

The chitosan suspension (2% *w*/*v*) was treated in the DBD device with filling helium of 5 L/min and was cooled for 2 min after every 3 min of treatment. The treated chitosan was fully mixed with glutaraldehyde solution (2% *w*/*v*) in the ratio of 4:1 *v*/*v*. Then, the mixed solution stood in a constant temperature water of 50 °C for 1 h, bath and at room temperature for 48 h in sequence. The effect of different discharge voltages and time on chitosan dissolution was studied, which indicated that the treatment at 60 kV and 150 W for 15 min were the best parameters for the dissolution of chitosan. Therefore, this parameter combination would be used for the subsequent plasma treatment of chitosan suspension.

#### 3.3.2. Sodium Alginate Hydrogel

Sodium alginate solution (2% *w*/*v*) was placed in the DBD system and treated for 5 min. Helium was introduced at a rate of 5 L/min. Next, a 1/4 volume of sodium alginate solution (1% *w*/*v*) was added into the solution, drop by drop. Then, the mixed solution stood in a constant temperature water bath of 60 °C for 1 h and at room temperature for 24 h in sequence.

#### 3.3.3. Chitosan-Sodium Alginate Composite Hydrogel

The plasma-treated chitosan suspension (2% *w*/*v*) was mixed with the sodium alginate solution (2% *w*/*v*) in the ratio of 1:1 *v*/*v* and processed in an ultrasonic oscillator at 70% power for 20 min at 50 °C. Then, the mixture was immersed in an acetic acid solution (2% *w*/*v*) and stood at room temperature for 48 h.

### 3.4. Lyophilisation

The prepared hydrogels were washed repeatedly in deionized water until pH neutral and soaked in deionized water for 3 days (with changing water every 8 h to remove residual substances). The obtained hydrogel was cut into several 1 cm^3^ cubes and divided into two groups. One group was put into a freeze-dryer (LC-10N-50A, LICHEN, Shanghai, China) for 48 h to obtain freeze-dried hydrogel, the other group was soaked in PBS buffer at pH = 7.4.

### 3.5. Characterization of Hydrogels

#### 3.5.1. Surface Morphology of Hydrogel Sections

Freeze-dried hydrogels were brittlely fractured under liquid nitrogen and then treated with gold spraying using an ion sputtering apparatus for 120 s. Then, the surface morphology of the sprayed sections was observed using the SEM (Sigma500, ZEISS, Gottingen, Germany).

#### 3.5.2. FTIR of Hydrogels

The freeze-dried hydrogels were separately ground into a powder and pressed into tablets, and the corresponding FTIR were measured in 400–4000 cm^−1^ using the spectrometer (PerkinElmer Spectrum Two, PerkinElmer, Waltham, MA, USA).

#### 3.5.3. Water Absorption and Swelling Ratio of Hydrogels

Using an electronic balance, we weighed the freeze-dried hydrogel as $W_{0}$. Then, we put the freeze-dried hydrogel strips in deionized water in a constant temperature incubator at 25 °C, removed the hydrogel from the deionized water every hour, wiped off the water on the surface of the hydrogel and weighed it as $W_{n}$. The formula for calculating the water absorption and swelling ratio of the hydrogel is as follows.
(1)$$Water\;Absorption=\frac{W_{n}-W_{0}}{W_{n}}\times 100\%$$
(2)$$Swelling\;Ratio=\frac{W_{n}-W_{0}}{W_{0}}\times 100\%$$

#### 3.5.4. Porosity of the Hydrogels

Using an electronic balance to weigh the freeze-dried hydrogel as W, we immersed the freeze-dried hydrogel in anhydrous ethanol for 1 h until it is full of ethanol, then wiped off the ethanol on the surface of the hydrogel and weighed it as $W_{1}$. We pressed the hydrogel that had been full of anhydrous ethanol into the beaker, which was full of anhydrous ethanol and weighed the mass of overflowing anhydrous ethanol as $W_{2}$. The porosity of the freeze-dried hydrogel (the volume of the pore space in the proportion of the total volume of the hydrogel) is as follows.
(3)$$Porosity=\frac{W_{1}-W}{W_{2}}\times 100\%$$

#### 3.5.5. Mechanical Properties of Hydrogels

The axial compression experiment at 200 N with the compression rate of 1 mm/min was carried out on the prepared hydrogel using a universal mechanical testing machine (CMT6104, MTS, Eden Prairie, MN, USA), and the compression modulus of the hydrogel was calculated by the stress–strain curve. The calculation formula is as follows.
(4)$$E=\frac{\sigma }{\varepsilon }$$
where σ is stress (kPa) and ε is the strain (%).

#### 3.5.6. Degradation of Hydrogels In Vitro

The compressive modulus of the hydrogels was recorded as E_0_ when they were not degraded. Then, the hydrogels were immersed in a container with PBS buffer of pH = 7.4, placed in a constant-temperature shaking table of 37 °C with the rotational speed of 30 r/min to simulate the vivo degradation process. The compressive modulus of the hydrogels was recorded as E when the hydrogels were taken out at fixed intervals of every 2 days and E/E_0_ was used to characterize the degradation property.

### 3.6. Methods of Statistical Analysis

All quantitative values are presented as mean ± standard deviation. All experiments were performed using at least three replicates. The data were plotted using Origin 2021. TOPSIS is an effective multi-indicator evaluation method, by which the optimal and worst values for each indicator can be obtained from all the values of each indicator and then the distances from the point corresponding to each evaluation value to the optimal and worst points are calculated to obtain the overall evaluation value (represented by *f* in this paper). The larger of the overall evaluation value means the better overall performance.

## 4. Conclusions

This study shows that plasma treatment can fully dissolve chitosan and prepare its aqueous solution. Meanwhile, in this experiment, chitosan hydrogel, sodium alginate hydrogel and chitosan-sodium alginate composite hydrogel with and without plasma assistance were prepared. SEM and universal mechanical testing machine were used to characterize the surface morphology and mechanical properties. In addition, the swelling ratio, water absorption, porosity and degradation ratio of the hydrogels were also investigated. The results show that the mechanical strength and anti-degradation properties of the hydrogels with plasma-assisted preparation are significantly improved. Meanwhile, the porosity, water absorption, and swelling are slightly decreased. The comparison of the overall evaluation value shows that the overall performance of plasma-assisted hydrogel is better than that of the hydrogel without plasma. In the six groups of hydrogels, the composite hydrogel of chitosan and sodium alginate prepared by plasma-assisted method has the best comprehensive performance with the compression modulus of 8.82 kPa and 46.28% of the compression modulus is retained on the 16th day of the simulated degradation, showing the strong mechanical and anti-degradation properties. This study provides an environmentally friendly and non-toxic hydrogel preparation method using plasma treatment and does not require traditional toxic cross-linking agents. The plasma prepared composite hydrogels show significant advantages in overall performance. In the future, it will be a potential research direction to combine other materials like gelatin hydrocolloids [49] and custom-designed nanoparticles [50], which will significantly improve the properties of hydrogels in special fields, such as wound healing, combined with antibacterial infection and water retention. This study will provide a new strategy for the design of functional hydrogels.

## Figures and Tables

**Figure 1 ijms-25-02418-f001:**
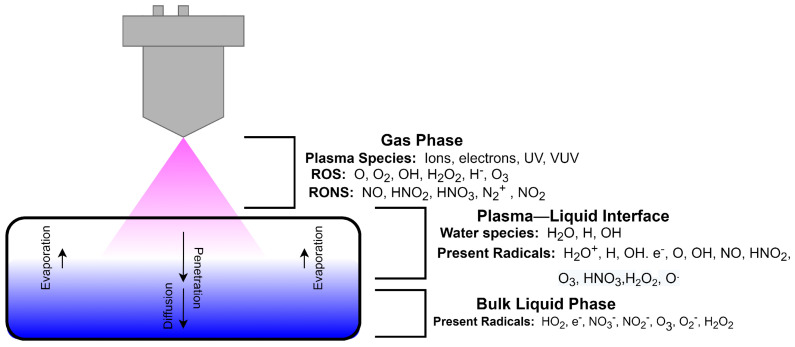
Schematic diagram of plasma water treatment [23].

**Figure 2 ijms-25-02418-f002:**
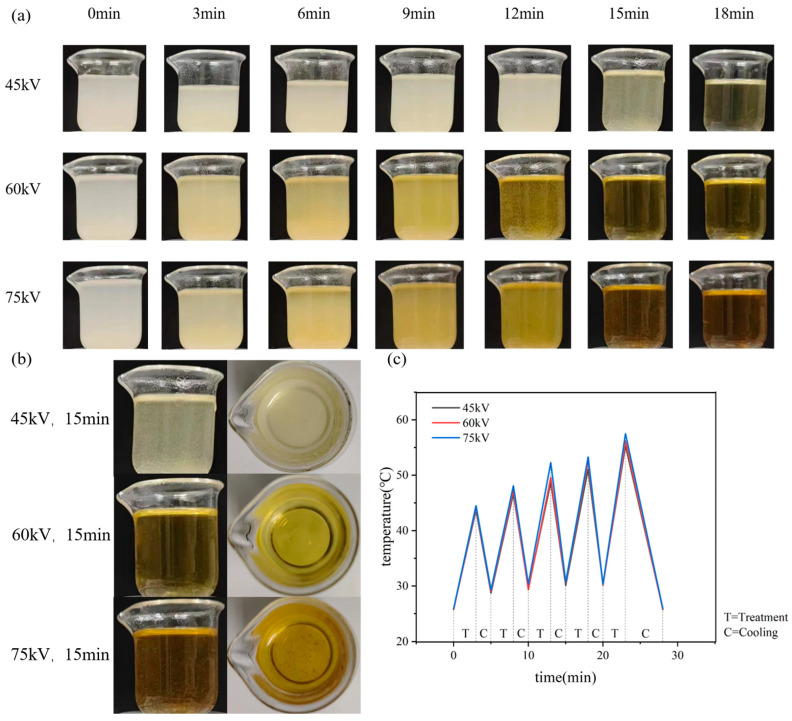
Aqueous chitosan solution treated with different plasma parameters. (**a**) the dissolution process of chitosan at different discharge voltages and time; (**b**) the dissolution situation of chitosan with 15 min treatment at different discharge voltages; (**c**) temperature changing during chitosan treatment and cooling process.

**Figure 3 ijms-25-02418-f003:**
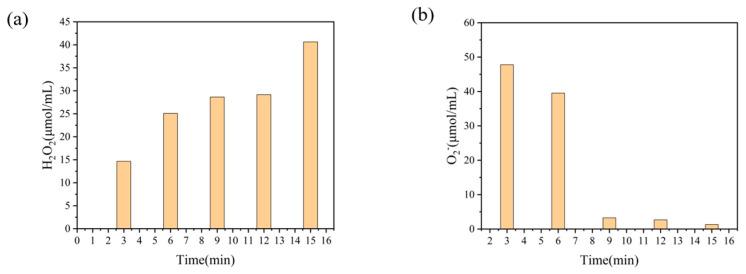
Changes in active substance content during plasma treatment.

**Figure 4 ijms-25-02418-f004:**
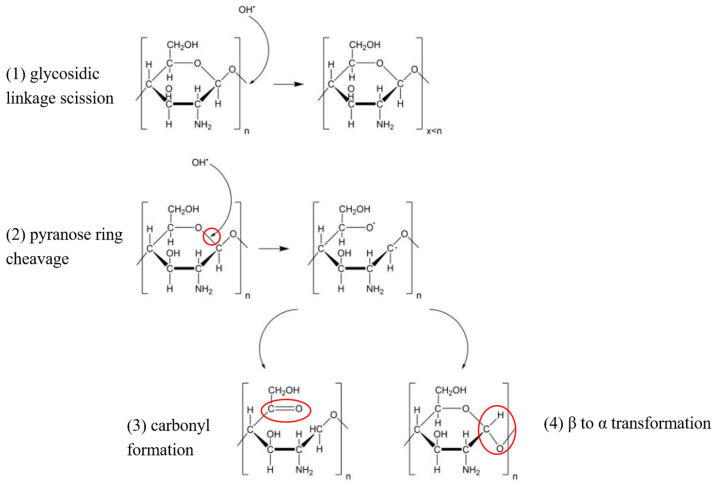
Possible chemical pathways of chitosan modification by hydroxyl radicals generated during plasma treatment of solutions in atmospheric pressure. The red circles represent the C-O bond position of the pyranose ring attacked by hydroxyl radicals, as well as the formation of a carbonyl group and conformational change (β → α).

**Figure 5 ijms-25-02418-f005:**
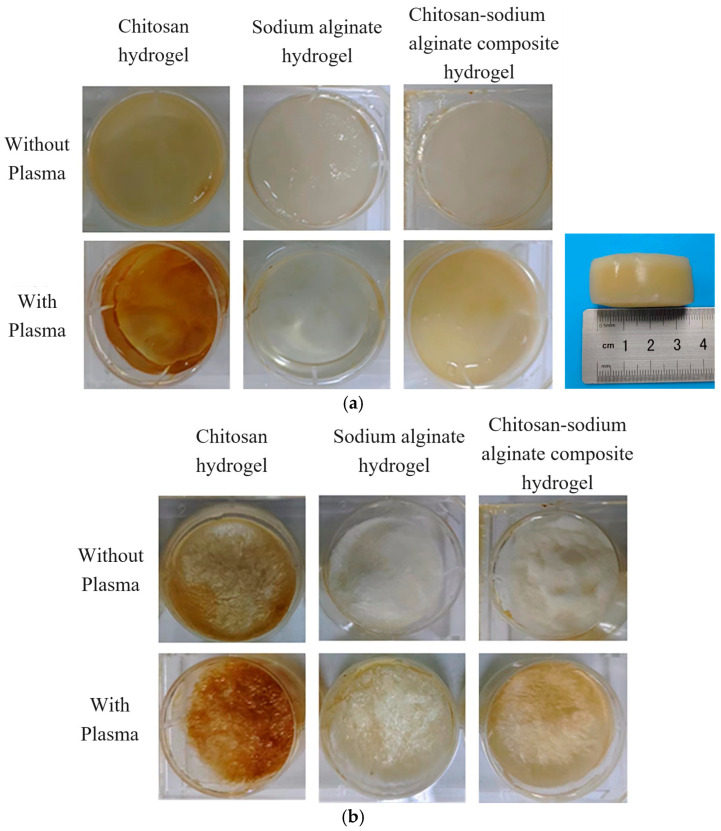
Comparison of three hydrogels with and without plasma treatment and their effects after freeze-drying. (**a**) Comparison of three kinds of hydrogels with and without plasma treatment (the subfigure with the ruler shows the sideview of chitosan-sodium alginate composite hydrogel with plasma). (**b**) Comparison of hydrogels with and without plasma treatment after freeze-drying.

**Figure 6 ijms-25-02418-f006:**
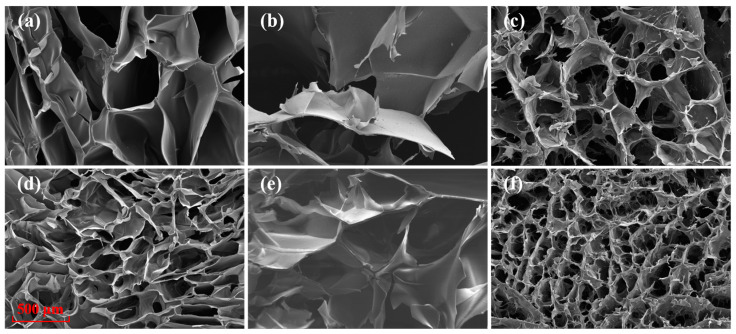
Surface morphology of the hydrogels. (**a**–**c**) are chitosan hydrogel, sodium alginate hydrogel and chitosan-sodium alginate composite hydrogel, and (**d**–**f**) are chitosan (DBD) hydrogel, sodium alginate (DBD) hydrogel, and chitosan (DBD)-sodium alginate composite hydrogel.

**Figure 7 ijms-25-02418-f007:**
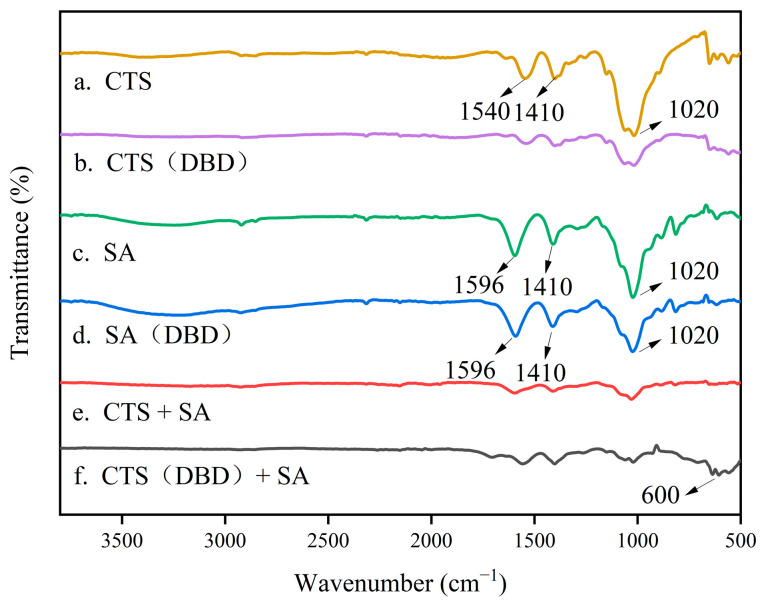
Comparison of the FTIR spectra of chitosan hydrogel, sodium alginate hydrogel, chitosan-sodium alginate composite hydrogel without (**a**,**c**,**e**) and with (**b**,**d**,**f**) plasma treatment.

**Figure 8 ijms-25-02418-f008:**
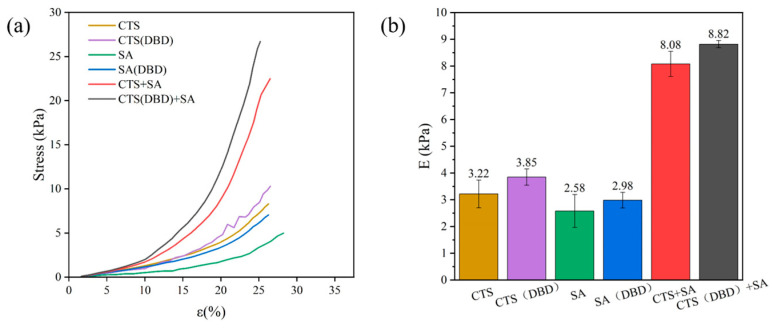
Stress–strain curves and compression modulus of different types of hydrogels, which are chitosan hydrogel, sodium alginate hydrogel, chitosan-sodium alginate composite hydrogel without and with plasma treatment. (**a**) Stress–strain curves for different types of hydrogels; (**b**) compression modulus of different types of hydrogels.

**Figure 9 ijms-25-02418-f009:**
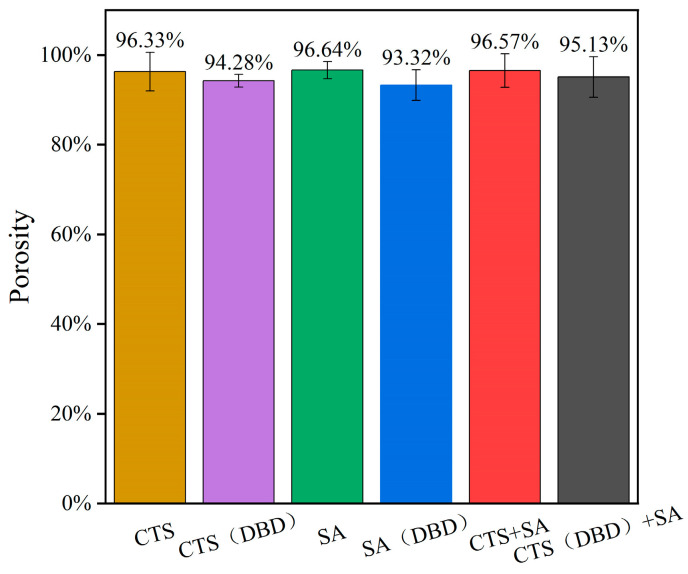
Porosity of different types of hydrogels, which are chitosan hydrogel, sodium alginate hydrogel, chitosan-sodium alginate composite hydrogel without and with plasma treatment.

**Figure 10 ijms-25-02418-f010:**
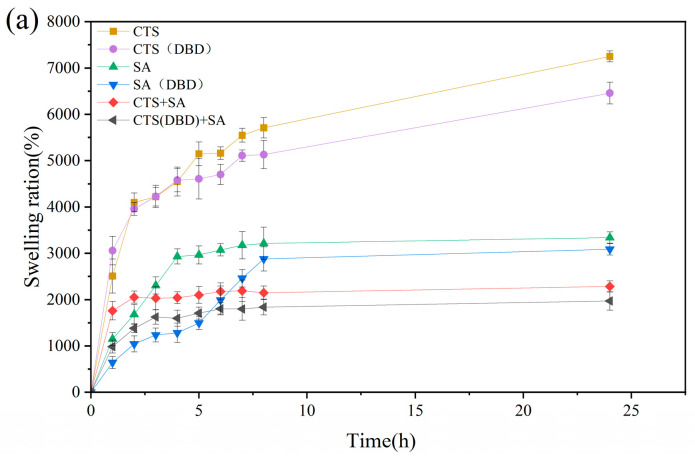
Swelling process of different types of hydrogels, which are chitosan hydrogel, sodium alginate hydrogel, chitosan-sodium alginate composite hydrogel without and with plasma treatment. (**a**) Swelling equilibrium curves for different types of hydrogels; (**b**) swelling ratios and (**c**) water absorption at swelling equilibrium.

**Figure 11 ijms-25-02418-f011:**
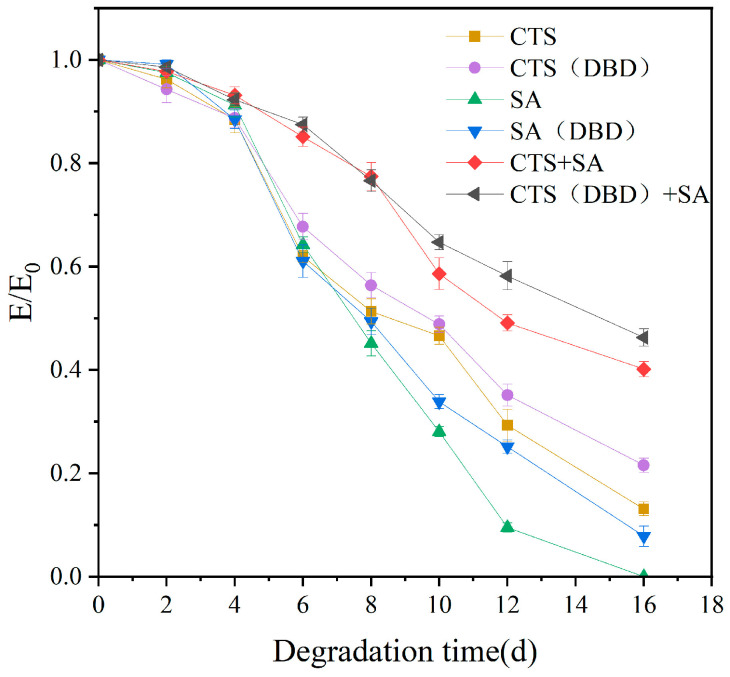
Degradation properties of different types of hydrogels, which are chitosan hydrogel, sodium alginate hydrogel, chitosan-sodium alginate composite hydrogel without and with plasma treatment.

**Figure 12 ijms-25-02418-f012:**
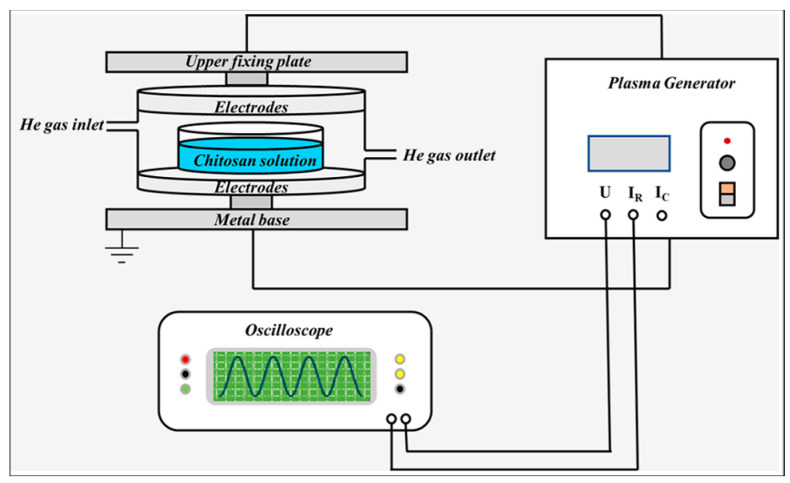
Experimental setup.

**Figure 13 ijms-25-02418-f013:**
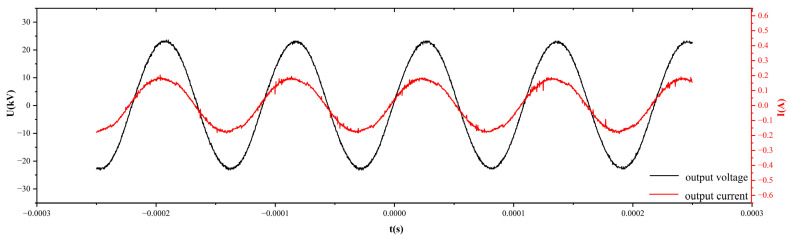
Voltage and current waveforms of helium discharge at 60 kV.

**Figure 14 ijms-25-02418-f014:**
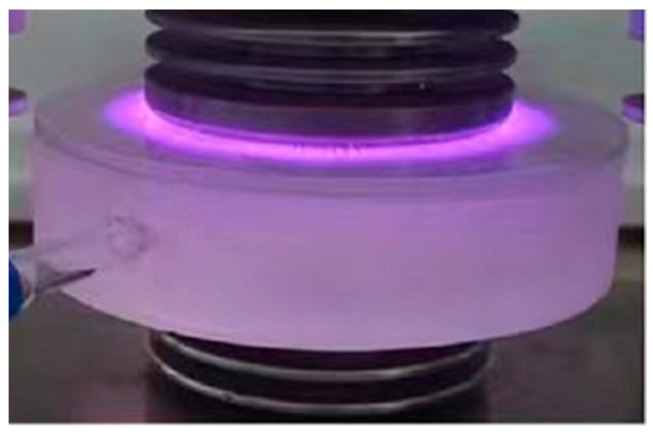
Discharge image of helium filled into the chamber at atmospheric pressure.

**Table 1 ijms-25-02418-t001:** Preparation parameters of chitosan hydrogel, sodium alginate hydrogel, and chitosan-sodium alginate composite hydrogel.

Hydrogel Name	Formula
Chitosan hydrogel	2% chitosan (dissolved in acetic acid) + glutaraldehyde
	2% chitosan (DBD) + glutaraldehyde
Sodium alginate hydrogel	2% sodium alginate + calcium chloride
	2% sodium alginate (DBD) + calcium chloride
Chitosan-sodium alginatecomposite hydrogel	2% chitosan (dissolved in acetic acid) + 2% sodium alginate
	2% chitosan (DBD) + 2% sodium alginate

**Table 2 ijms-25-02418-t002:** Comparison of properties of different hydrogels.

Group	Compression Modulus (kPa)	Porosity (%)	Water Absorption Ratio (%)	Swelling Ratio (%)	E/E_0_ (%)	*f*-Value
CTS	3.22	96.33	98.64	7248.60	13.12	0.206
CTS (DBD)	3.85	94.28	98.47	6456.73	21.56	0.335
SA	2.58	96.64	97.09	3339.09	0.00	0.184
SA (DBD)	2.98	93.32	96.86	3086.00	7.84	0.253
CTS + SA	8.08	96.57	95.81	2284.21	40.16	0.856
CTS (DBD) + SA	8.82	95.13	95.17	1969.84	46.28	0.983

## Data Availability

The data presented in this study are available on request from the corresponding author. The data are not publicly available yet because funded grants are still ongoing.

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
