# Peer review of "Dielectric Barrier Discharge Plasma-Assisted Preparation of Chitosan-Based Hydrogels"

_ijms, 2024, doi:10.3390/ijms25042418_

Round 1
Reviewer 1 Report
Comments and Suggestions for Authors
Reviewer 2 Report
Comments and Suggestions for Authors
The manuscript titled “Dielectric Barrier Discharge Plasma-Assisted Preparation of Chitosan-Based Hydrogels” by Liang, R.; et al. is a scientific work where the authors assessed the impact of plasma exposure on chitosan hydrogel physico-chemical properties. For it, the authors exploited many complementary techniques as scanning electron microscopy (morphology), fourier transform infrared spectroscopy (chemical analysis), tensile tests (mechanical properties), and the water uptake and durability of the prepared hydrogels. The achieved results can be interesting to design materials with improved properties for many applications. The manuscript is generally well-written.
However, it exists some points that need to be addressed (please, see them below detailed point-by-point) to improve the scientifc quality of the submitted manuscript paper before this article will be consider for its publication in the International Journal of Molecular Sciences.
1) KEYWORDS: The authors should consider to add the term “dielectric barrier” in the keyword list.
2) INTRODUCTION. “Chitosan is promising for (…) drug delivery [Error! Reference source not found.]” (lines 27-29). Please, the authors should fix this issue. This comment should be taken into account for the rest of the main manuscript body text.
3) “CO2 activation (…) wastewater treatment” (lines 34-36). The full-name of the chemical compound should be added and then, the chemical formula placed between brackets. This point shoud be covered for the rest of the manuscript text.
4) MATERIALS AND METHODS. “2.1. Experimental Materials and Instruments” (lines 106-113). The name and country details of the supplier manufacturer should be furnished for all the chemicals and techniques used in this research. This comment is extrapolated for the rest of this section.
5) “2.5.5. Mechanical properties of the hydrogels” (lines 189-192). The physical formula to ascertain the Young’s modulus values of the examined chitosan hydrogels should be added (eventually a relevant reference citation is welcome).
6) RESULTS AND DISCUSSION. Figure 6 (line 276). The standard deviation (SD) bars should be added for each tested condition. Then, further statistical analysis like the Student’s t-test or the analysis of variance (ANOVA) needs to be devoted to discern if the observed differences among the acquired data is statistically relevant. Same comment for the Figure 11, panel b (line 415), Figure 12 (line 435) and Figure 13, panels b and c (line 477).
7) Figure 9 (line 355). The lateral scale bar dimensions should be provided in the respective figure caption (it is not possible to visualize this information in each scanning electron microscopy image. Then, did the authors observe any hydrogel degradation effect caused by the plasma irradiation? Some information should be furnished in this regard.
8) “Figure 11 (…) compressive modulus of chitosan and sodium algniate composite hydrogel (8.08 kPa) (…) chitosan hydrogel (3.22 kPa) and sodium alginate hydrogel (3.85 kPa)” (lines 400-402). Similar comment than the aboved described. The authors should provide data information about the SD bars (which has been already implemented in the panel b of this Fig).
9) “2.7. Porosity of Different Types of Hydrogels Prepared by Plasma. (…) As can be seen than the porosities of these hydrogels were between 93 and 97%” (lines 420-425). Do the authors consider these observed differences in terms of porosity significant to achieve different mechanical and water uptake hydrogel performances?
10) CONCLUSION. This section clearly outlines the most relevant outcomes found in this work. The authors should discuss about some potential future action lines to pursue this research like the potential combination of the methodology developed by the authors with other materials like gelatin hydrocolloids [1] or custom-designed nanoparticles [2] to develop suitable platforms for regenerative biomedicine.
[1] Pele, K.G.; et al. Hydrocolloids of Egg White and Gelatin as Platform for Hydrogel-Based Tissue Engineering. Gels 2023, 9, 505. https://doi.org/10.3390/gels9060505.
[2] Sutthavas, P.; et al. Zn-Loaded and Calcium Phosphate-Coated Degradable Silica Nanoparticles Can Effectively Promote Osteogenesis in Human Mesenchymal Stem Cells. Nanomaterials 2022, 12, 2918. https://doi.org/10.3390/nano12172918.
Finally, the citation references are in the proper format of the International Journal of Molecular Sciences (No actions are requested from the authors).
Round 2
Reviewer 2 Report
Comments and Suggestions for Authors
The authors did a great deal of effort to cover all the suggestions raised by the Reviewers. Due to it, the scientific quality of the manuscript was greatly improved.
It lacks the citation to the combination with other materials as hydrocolloids [1] or tailored nanoparticles [2] to design promising platforms for biomedicine purposes:
[1] Pele, K.G.; et al. Hydrocolloids of Egg White and Gelatin as Platform for Hydrogel-Based Tissue Engineering. Gels 2023, 9, 505. https://doi.org/10.3390/gels9060505.
[2] Sutthavas, P.; et al. Zn-Loaded and Calcium Phosphate-Coated Degradable Silica Nanoparticles Can Effectively Promote Osteogenesis in Human Mesenchymal Stem Cells. Nanomaterials 2022, 12, 2918. https://doi.org/10.3390/nano12172918.
